# Proeryptotic Activity of 4-Hydroxynonenal: A New Potential Physiopathological Role for Lipid Peroxidation Products

**DOI:** 10.3390/biom10050770

**Published:** 2020-05-16

**Authors:** Mario Allegra, Ignazio Restivo, Alberto Fucarino, Alessandro Pitruzzella, Sonya Vasto, Maria Antonia Livrea, Luisa Tesoriere, Alessandro Attanzio

**Affiliations:** 1Dipartimento di Scienze e Tecnologie Biologiche, Chimiche e Farmaceutiche, Università di Palermo, 90123 Palermo, Italy; mario.allegra@unipa.it (M.A.); ignazio.restivo@unipa.it (I.R.); sonya.vasto@unipa.it (S.V.); maria.livrea@unipa.it (M.A.L.); alessandro.attanzio@unipa.it (A.A.); 2Dipartimento di Biomedicina, Neuroscienze e Diagnostica Avanzata, Università di Palermo, 90127 Palermo, Italy; alberto.fucarino@unipa.it (A.F.); alessandro.pitruzzella@unipa.it (A.P.); 3Consorzio Universitario di Caltanissetta, Università di Palermo, 90127 Palermo, Italy

**Keywords:** 4-hydroxynonenal, lipid peroxidation products, eryptosis, prostaglandins, RBC, inflammation

## Abstract

Background: Eryptosis is a physiological, apoptosis-like death of injured erythrocytes crucial to prevent premature haemolysis and the pathological sequalae generated by cell-free haemoglobin. When dysregulated, the process is associated to several inflammatory-based pathologies. 4-Hydroxy-trans-2-nonenal (HNE) is an endogenous signalling molecule at physiological levels and, at higher concentrations, is involved in the pathogenesis of several inflammatory-based diseases. This work evaluated whether HNE could induce eryptosis in human erythrocytes. Methods: Measurements of phosphatidylserine, cell volume, intracellular oxidants, Ca^++^, glutathione, ICAM-1, and ceramide were assessed by flow cytometry. Scanning electron microscopy evaluated morphological alterations of erythrocytes. Western blotting assessed caspases. PGE_2_ was measured by ELISA. Adhesion of erythrocytes on endothelial cells was evaluated by gravity adherence assay. Results: HNE in the concentration range between 10–100 µM induces eryptosis, morphological alterations correlated to caspase-3 activation, and increased Ca^++^ levels. The process is not mediated by redox-dependent mechanisms; rather, it strongly depends on PGE_2_ and ceramide. Interestingly, HNE induces significant increase of erythrocytes adhesion to endothelial cells (ECs) that are in turn dysfunctionated as evident by overexpression of ICAM-1. Conclusions: Our results unveil a new physiopathological role for HNE, provide mechanistic details of the HNE-induced eryptosis, and suggest a novel mechanism through which HNE could exert pro-inflammatory effects.

## 1. Introduction

Eryptosis is a physiological, apoptosis-like, suicidal death of erythrocytes (RBC) [1]. Hallmark of the eryptotic process is the loss of membrane phospholipid asymmetry due to the translocation of the cell membrane phospholipid phosphatidylserine (PS) from the inner to the outer leaflet by Ca^++^-sensitive scramblase [2]. This process allows eryptotic RBCs to be recognized by macrophages, phagocytosed, degraded, and cleared from circulation.

Utterly distinct from the erythrocytic senescence, eryptosis ostensibly serves the same purpose as apoptosis, i.e., the disposal of defective cells without breaking cell membrane integrity and releasing eventual pro-inflammatory, cytosolic material to the extracellular environment. Eryptosis can be, therefore, envisaged as a preemptive measure of the body to reduce premature haemolysis of injured RBC, thus preventing the detrimental increase of cell-free haemoglobin in circulation [3].

Excessive eryptosis, however, has been implicated in a number of pathological conditions such as anaemia, renal and hepatic failure, sepsis, and malignancy [3,4,5,6]. Moreover, due to propensity of eryptotic RBC to adhere to endothelial cells (ECs), eryptosis has been involved in the pathogenesis of inflammatory and cardiometabolic diseases [3,4].

Eryptosis is triggered by several pathophysiological cell stressors and orchestrated by a complex machinery of lipids, membrane proteins, ion channels, and cellular enzymes. In this regard, the increase of endocellular Ca^++^ levels play a key role within the eryptotic process. It has, indeed, been shown that several proeryptotic stimuli can activate phospholipase A_2_ (PLA_2_) that releases arachidonic acid (AA), leading to prostaglandin endoperoxide synthase (PGHS) activation. The resulting increase of PGE_2_ levels then activates and opens nonselective cationic membrane (NSCM) channels, leading to an increment of Ca^++^ endocellular levels that in turn activates several molecular targets including Ca^++^-dependent K^+^ channels. The resulting membrane hyperpolarization increases the electrical driving force for Cl^−^ exit, and the resulting cellular loss of KCl, with osmotically obliged water efflux, determines RBC shrinkage [6]. On the other hand, Ca^++^ influx, together with platelet activating factor (PAF), activates membrane sphingomyelinase (SMase) responsible for the increase of ceramide levels that plays a key role in the apoptotic process [3,4]. Indeed, Ca^++^ and ceramide are able to respectively activate scramblase and to inhibit flippases, involved in PS externalization. Finally, increase of Ca^++^ levels also activate cysteine-calpain endopeptidases responsible for the membrane blebbing that increases the adhesion capacity of RBC [7].

4-Hydroxy-*trans*-2-nonenal (HNE) is an α,β-unsaturated aldehyde endogenously generated by the radical-mediated peroxidation of ω-6 polyunsaturated fatty acids. Its reactive carbonyl moieties, together with the conjugated double bonds, generate two electrophilic centres (the carbonyl carbon atom and the β-carbon one), both of which can condense with suitable nucleophilic moieties of biological targets, such as proteins, DNA, and phospholipids [8,9]. Thanks to its ability to covalently modify biomolecules, HNE modulates many physiopathological processes [10] and its role highly depends on its concentrations. Indeed, if at physiological (74 nM to 1 µM) or at low/sublethal plasma concentrations (up to 10 µM) HNE acts as an endogenous signalling molecule, higher and toxic plasma levels (20 µM to 5 mM) are involved in the onset and propagation of several human diseases [8,9,11,12,13,14,15,16,17]. As a signalling molecule, HNE plays a key role in several regulatory mechanisms responsible for glutathione biosynthesis, insulin secretion, and stimulation of antioxidant enzymes [10,18]. On the other hand, toxic concentrations of HNE constitute one of the main driving forces in the development and progression of several inflammatory-based conditions [19,20,21]. Among these, atherosclerosis, neurological diseases, and metabolic syndrome risk factors (dyslipidaemias, insulin resistance, and obesity) have all been recently correlated to high HNE plasma levels in the range between 10 and 100 µM [22]. Covalent protein modifications, enzyme activity, and gene expression alterations by HNE are involved in the mechanisms underlying the above-cited pathological conditions.

In light of the mechanisms orchestrating the eryptotic process, strongly regarding membrane structure and function and considering the key role of HNE in membrane alterations supporting inflammatory-based pathologies [11], we here evaluated whether HNE at physiopathological concentrations could induce eryptosis in human RBCs. Mechanistic insights for the observed proeryptotic activity of HNE were investigated, and pathological relevance of the observed results were discussed.

## 2. Materials and Methods

Unless otherwise specified, all reagents and chemicals were from Sigma-Aldrich (Milan, Italy) and of the highest purity grade available.

### 2.1. Red Blood Cells and Treatment

Blood was collected from healthy volunteers (n = 5; age range between 23 and 65 years with physiological body mass index) with informed consent, and RBCs were immediately isolated by 20 min centrifugation at 2000× *g* at 4 °C over Ficoll (Biochrom KG, Berlin, Germany) gradient. RBCs (0.4% haematocrit) were incubated at 37 °C, 5% CO_2_, and 95% humidity in Ringer solution containing (mM) 125 NaCl, 5 KCl, 1 MgSO_4_, 32 *N*-2-hydroxyethylpiperazine-*N*-2-ethanesulfonic acid (HEPES)/NaOH, 5 Glucose, and 1 CaCl_2_ at pH 7.4 for the indicated time.

RBCs were treated for 24 h with HNE and were added in Ringer’s solution at a final 0.1% (*v*/*v*) ethanol (EtOH) concentration. Preliminary experiments showed that, under these conditions, EtOH did not have any effect on cells and therefore control RBCs were incubated with EtOH.

In selected experiments, RBCs either were incubated in Ringer’s solution in the absence of CaCl_2_ and then treated with HNE or were preincubated for 1 h in Ringer’s solution in the presence of 50 µM Acetylsalicylic Acid (ASA) diluted in EtOH.

The recruitment of the healthy volunteers was conducted within the project “Discovery of molecular and genetic/epigenetic signatures underlying resistance to age-related diseases and comorbidities (DESIGN)” Bando 2015 Prot 20157ATSLF, funded by the Italian Ministry of Education, University, and Research. The study protocol was approved by the Ethic Committee of Palermo University Hospital (Nutrition and Longevity, No. 032017) The study was conducted in accordance with the Declaration of Helsinki and its amendments.

### 2.2. Flow Cytometry

#### 2.2.1. Measurement of Phosphatidylserine (PS) Externalization and Forward Scatter (FSC)

RBCs were washed once in Ringer’s solution, pH 7.4, and adjusted at 1.0 × 10^6^ cells/mL with a binding buffer according to the manufacturer’s instructions (eBioscience Inc., San Diego, CA, USA). In experiments designed to evaluate PS externalization, cell suspension (100 μL) was incubated with 5 μL Annexin V-FITC at room temperature in the dark for 15 min. Samples of at least 1 × 10^4^ cells were then analysed by flow cytometry (FACS) for both FSC and Annexin V-fluorescence intensity (the latter in fluorescence channel FL-1, with an excitation wavelength of 488 nm and an emission wavelength of 530 nm), with an Epics XL™ flow cytometer, using Expo32 software, version 1.1C (Beckman Coulter, Fullerton, CA).

#### 2.2.2. Measurement of Intracellular Reactive Oxygen and Nitrogen Species

Level of Reactive Oxygen and Nitrogen Species (RONS) was monitored by measuring the fluorescence changes resulting from oxidation of dichloro-dihydro-fluorescein diacetate (DCF-DA). Briefly, 10 μM fluorescence probe was added to 1.0 × 10^5^ cells, 30 min before the end of the treatment, in the dark. Cells were collected by centrifugation (2000× *g*, 4 °C, 5 min), washed, resuspended in PBS, and analysed as above reported.

#### 2.2.3. Measurement of Cytosolic Ca^++^

Levels of intracellular Ca^++^ was monitored by measuring the fluorescence changes of the cell-permeable dye fluo-3 AM resulting from its binding with Ca^++^. Briefly, 10 μM fluorescence probe was added to 1.0 × 10^5^ cells 30 min before the end of the treatment in the dark. Cells were collected by centrifugation (2000× *g*, 4 °C, 5 min), washed, resuspended in PBS, and analysed as above reported.

#### 2.2.4. Measurement of Ceramide

Level of ceramide was monitored by flow cytometry as follows. Briefly, 24 h after the treatment, 1.0 × 10^5^ cells were incubated with 1 µg of a mouse antihuman, ceramide monoclonal antibody (Alexis, Gruenberg, Germany) in PBS containing 0.1% bovine serum albumin (BSA) for 1 h at 37 °C. Cells were then washed twice with PBS-BSA and incubated for 30 min with 20 µL of a goat anti-mouse, polyclonal, fluorescein isothiocyanate-conjugated, secondary antibody (Pharmingen, Germany) diluted 1/50 in PBS-BSA. Cells were then collected by centrifugation (2000× *g*, 4 °C, 5 min), washed, resuspended in PBS, and analysed as above reported.

### 2.3. Haemolysis

After 24 h treatment, a volume (0.2 mL) of the incubation mixture was diluted with 10 volumes of PBS and centrifuged at 1000× *g* for 10 min to precipitate the cells. The absorbance of the supernatant was then evaluated at 408 nm (DU-640 Spectrophotometer Beckman, Brea, CA, USA). Similarly, a volume of the incubation mixture was treated with 10 volumes of 5 mM sodium phosphate buffer, pH 7.4 (hypotonic PBS), and briefly exposed to an ultrasonic bath. The absorbance value at 408 nm was taken as 100% haemolysis.

### 2.4. Western Blotting

RBCs were washed twice with PBS, resuspended in lysis buffer (20 mM Tris-HCl, pH 7.6, 100 mM NaCl, 10 mM MgCl_2_, 2 mM PMSF, 0.5 mM DTT, and 2 mg/mL Lysozyme) with protease inhibitor cocktail (Roche Applied Science, Indianapolis, IN, USA), and sonicated for 60 s on ice with Labsonic LBS1-10 (Falc Instruments, Treviglio, Italy). After centrifugation at 40,000× *g* for 1 h at 4 °C, supernatant was collected. Protein concentration of each sample was determined by using Bradford protein assay reagent (Bio-Rad, Milan, Italy). Protein samples (50 μg/line) were separated on 10% SDS-PAGE and transferred to nitrocellulose membrane. Immunoblots were blocked overnight at 4 °C with 5% skimmed milk, followed by incubation with a 1:1000 dilution of mouse antihuman, caspase-3 monoclonal antibody (clone 3C119, Santa Cruz Biotechnology, Milan, Italy) for 1 h at room temperature. Blots were washed two times with Tween 20/Tris-buffered saline and incubated with a 1:2000 dilution of rabbit anti-mouse, horseradish peroxidase-conjugated anti-IgG antibody (Dako, Santa Clara, CA, USA) for 1 h at room temperature. Blots were then washed five times with Tween 20/Tris-buffered saline and then developed by enhanced chemiluminescence (Amersham Life Science, Milan, Italy).

### 2.5. Scanning Electron Microscopy (SEM)

SEM is a microscopic technique that does not use photons of light as optical microscopy but an electron beam that hits the sample. Thanks to the much lower electron wavelength, compared to the photon one, the resolution power of a scanning electron microscope is significantly higher than a visual display on an optical microscope. We are thus able to obtain images with magnification up to 20,000×, in which the details of the eryptotic erythrocytes are very evident. After 24 h of treatment, RBCs were prefixed for about 60 min in a 2% paraformaldehyde and 0.5% glutaraldehyde solution in Milloning buffer pH 7.3. The samples were then washed once in the same buffer and postfixed with 1% osmium tetraoxide in a Milloning buffer for two hours under stirring condition. Subsequently, the samples were washed again in a Milloning buffer and then dehydrated by an ascending ethanol scale (30%, 50%, 70%, 95%, and 100%): each step was repeated twice for 10 min. After the last passage in 100% ethanol, the samples were treated with HMDS (hexamethyldisilazane) and left for 24 h in solution until complete evaporation. Finally, samples were grafted onto the supports for visualization by means of silver paste and subsequently metallized with gold and observed using a scanning electron microscope (JSM-6301F; JEOL Ltd., Japan). At each step, the samples were centrifuged at 1000× *g* to create a cell pellet.

### 2.6. Glutathione Measurements

Intracellular GSH/GSSG levels were measured by employing a glutathione colorimetric assay kit according to manufacturer’s instruction (Invitrogen, Milan, Italy).

### 2.7. PGE_2_ Measurements

After incubation, as indicated above, PGE_2_ released (pg/1 × 10^8^ cells) was quantified using a PGE_2_ Enzyme Immunoassay Kit (Cayman Chemical Corporation Inc., Milan, Italy) in accordance with the manufacturer’s protocol. Briefly, after treatment, cells were pelleted by centrifugation at 4 °C, for 5 min at 450× *g*. Samples of the supernatant were diluted at 1:2.5 with assay buffer. Then, a 100 µL sample, a 50 µL alkaline phosphatase PGE_2_ conjugate, and a 50 µL monoclonal anti-PGE_2_ EIA antibody were applied to a goat anti-mouse IgG-containing microtiter plate and incubated at room temperature for 2 h. After washing, 200 µL of *p*-nitrophenyl phosphate substrate solution was added and incubated at room temperature for 45 min. Finally, optical density at 405 nm was measured on a microplate reader. PGE_2_ concentrations in the samples were calculated from a PGE_2_ standard curve (25–5000 pg/mL) that was run in parallel.

### 2.8. Endothelial Cell Culture

#### 2.8.1. Adherence Assay of Eryptotic Rbcs

Human Umbilical Vein ECs (HUVEC) were purchased from PromoCell, Milan, Italy and used for up to four additional passages. Cells were grown in 75-cm^2^ flasks with endothelial cell basal medium MV2 (PromoCell) containing 100 pg/mL of heparin, 10% foetal bovine serum (GIBCO), 120 U/mL penicillin/streptomycin, and 30 pg/mL of endothelial cell growth supplement. At 85% confluence, cells were subcultured in 96-well, flat-bottom plates. Cellular monolayers were co-incubated with HNE-stimulated RBCs (25 or 50 µM) or control RBCs for 48 h. After treatment, the adherence of RBCs to HUVEC was measured by the gravity adherence assay as previously reported [23].

#### 2.8.2. Measurements of ICAM-1 Expression

After 24 h co-incubation with RBCs, HUVECs were washed with phosphate buffered saline (PBS), harvested with cell dissociation medium, and diluted in a washing buffer containing 0.1% bovine serum albumin and 1mM CaCl_2_. Aliquots (0.1 to 0.5 × 10^6^ cells in 20 μL) were incubated at 4 °C with 2 μg/well of a mouse antihuman ICAM-1 monoclonal antibody FITC-conjugated (Invitrogen, Milan, Italy). After 1 h at 4 °C, cells were washed and analysed by FACS.

### 2.9. Statistical Analysis

Results are expressed as mean ± SD of six independent experiments and conducted in triplicates. Statistical comparisons were performed by one-way analysis of variance (ANOVA) followed by Fisher’s correction for multiple comparisons using Prism version 8 (GraphPad Software Inc., San Diego, USA). In all cases, significance was accepted if the null hypothesis was rejected at the *p* < 0.05 level.

## 3. Results and Discussion

### 3.1. HNE Induces PS Externalization in Human RBCs

HNE, instead of a mere by-product of oxidative stress, is capable of modifying nucleophilic residues and, depending on its concentration, can either orchestrate cell signalling events or modify target proteins, thus dysregulating functional cellular processes such as proliferation and apoptosis [10,20,24,25,26,27,28,29,30,31,32]. In this paper, we focused on the HNE ability to modulate proapoptotic pathways and evaluated its eventual proeryptotic effects in a concentration range of pathophysiological relevance. As an initial approach toward this aim, HNE-induced PS externalized was assessed by incubating RBCs for 24 h in the absence (control) or in the presence of the aldehyde in the range between 10 and 100 µM. When compared to control cells, stimulation with HNE resulted in a significant, concentration-dependent increase of PS externalized expressed as % of Annexing V-binding cells. Indeed, the value progressively raised from 3.4% ± 0.2% with control cells to 61% ± 3% with 100 µM HNE (*p* < 0.05, Figure 1A,B). Successive experiments were, then, conducted with HNE between 25 and 50 µM, a concentration range frequently found in the plasma of obese subjects and patients affected by cardiometabolic and inflammatory pathologies [1,2,3,4]. Moreover, both in vitro and ex vivo evidences have reported that HNE concentrations close to 50 µM induce insulin resistance in muscle cells, in isolated skeletal muscle, and in adipocytes [14,33,34].

### 3.2. HNE Induces a Low Grade of Haemolysis in Human RBCs

In order to determine whether the HNE-mediated eryptosis was also associated to necrotic events, we next evaluated if and to what extent the aldehyde was able to induce haemolysis. To this end, RBCs were incubated for 24 h in the absence (control) or in the presence of HNE in the range between 25 and 50 µM and Hb release was measured. With respect to control cells, HNE induced a significant increase of haemoglobin release only at 50 µM with a value increasing from 3.8% ± 0.3% to 7.6% ± 0.4% (*p* < 0.05, Figure 1C).

When compared to the percentage of eryptotic cells, these values demonstrate that, in the process underlying HNE-induced RBC death, eryptotic mechanisms dominated over the necrotic ones. These results are also in line with those reported from our research group with other lipidic, proeryptotic stimuli such as oxysterols with which HNE seems to share the ability to stimulate predominantly eryptotic events [35]. Interestingly, present findings could also imply that, in vivo, HNE at physiopathological concentrations could eventually injure erythrocytes to an extent far from significantly inducing intravascular haemolysis and its detrimental clinical sequelae (gastrointestinal, cardiovascular, pulmonary, urogenital, and clotting disorders) [1,36,37].

### 3.3. HNE Induces Morphological Changes in Human RBCs

As before-mentioned, eryptosis is well known to induce evident morphological variations characterised by cell volume reduction and membrane blebbing. With the aim to investigate and characterise the HNE-induced morphological alterations in RBCs, we assessed FSC variations and performed scanning electron microscopy evaluations in both control and HNE-treated RBCs. As shown in Figure 2A, HNE determined a significant increase of FSC with respect to control cells with values increasing from 542 ± 2 Relative Units (RU) with untreated cells to 575 ± 3 and 600 ± 2 RU with 25 and 50 µM HNE respectively (*p* < 0.05). Our results suggest that HNE-induced eryptosis is accompanied with an apparent increase of RBC volume and differs from those reported in literature for other eryptotic models. It has indeed been demonstrated that RBCs, either stimulated with other eryptotic agents or isolated from patients suffering from sepsis or cardiometabolic or blood disorders, are characterised by a reduction of cell volume [1,4,6].

In order to further investigate these discrepancies, we next employed a SEM approach and analysed the morphological alteration in eryptotic RBCs. Figure 2 shows the onset after treatment with HNE for 24 h (Figure 2C,D) of altered cell morphologies not detectable in untreated samples (Figure 2B). Control RBCs have a shape and size comparable to the standards described in literature [38]. The acanthocytes (green arrow) appear characterized by a few irregular spicules with a blunt tip. These changes in their morphology are irreversible, while the echinocytes (red arrow), called porcupine cells or burr cells, show numerous narrow-tipped spicules regularly distributed on the surface of the red cell membrane. The latter alterations are instead considered reversible. The presence of numerous and irregular blebbing on the surface of eryptotic RBCs agrees with the apparent increase in volume detected by FSC and would exclude an actual increase of cellular volume.

By analyzing in details the complex cytological framework, primary echinocytes and polyhedrocytes were detected and identified (Figure 3B,C). These cellular forms are characteristic of an early eryptotic phase, during which the PS exposure process begins. Furthermore, the presence of agglutinated elements, with partial fusion of cell membranes, is of considerable interest (Figure 3D, fusion elements are indicated by the yellow arrows). These agglutination elements, through the interaction with other elements like platelets or ECs, can indeed sustain thrombotic and inflammatory processes usually associated to circulating eryptotic RBCs. From a mechanistic perspective, the different morphological changes found through SEM analysis could derive from the rearrangement of the membrane and cytoskeleton proteins [39]. In this regard, it is worthwhile to underline that HNE can accumulate in erythrocytic plasma membrane, can perturbate membrane fluidity, and can form protein adducts with spectrin and other membrane and cytoskeleton proteins in a variety of pathophysiological conditions (diabetes, renal disease, glucose-6-phosphate dehydrogenase deficiency, sickle cell anaemia, and tropical diseases) [40,41,42].

### 3.4. HNE Induces Cleavage of Procaspase 3 in Human RBCs

Notwithstanding that RBC is devoid of nucleus, mitochondria, and other organelles, it does have a molecular machinery, saved during the differentiation from erythroid progenitors, with functional caspases. Activation of these effectors is a key step in the eryptosis induced by either specific xenobiotic or physiopathological conditions [3,5,6,43]. Indeed, caspases often mediate eryptosis-triggered morphological modifications by degrading crucial proteins involved in the maintenance of erythrocyte’s shape and function, ultimately leading to an accelerated cell removal from circulation [39,44,45,46].

In light of the morphological changes triggered by HNE in eryptotic RBCs, we next evaluated the involvement of caspase-3 in the HNE-induced eryptosis. To this end, RBCs were incubated for 24 h in the absence (control) or in the presence of HNE in the range between 25 and 50 µM and caspase-3 cleavage was evaluated. With respect to control RBCs, stimulation with HNE at 25 and 50 µM significantly modified caspase-3 levels that increased from a control value of 0.12 ± 0.01 Arbitrary Units (AU) to 1.45 ± 0.2 and 1.73 ± 0.1 AU respectively. Conversely, procaspase-3 levels decreased from 1.13 ± 0.04 AU to 0.94 ± 0.05 and 0.70 ± 0.05 AU (*p* < 0.05, Figure 4).

These results are coherent with those previously reported from other groups, showing that caspase 3 is the principal mediator of HNE-mediated RBC damage during arsenic exposure [25] and is involved in the HNE-induced cell apoptosis in Jurkat cells [26].

### 3.5. Intracellular Ca^++^ Variations Are Involved in the HNE-Induced Eryptosis in Human Rbcs

As abovementioned, a wealth of studies confirmed a strict correlation between Ca^++^ increase and the activation of signalling processes that ultimately lead to activation of erythrocyte scramblase responsible for PS externalization [1,4,6]. In order to gain deeper insights on the mechanisms responsible for the HNE-induced eryptosis, we next evaluated the involvement of Ca^++^ in PS externalization. To this end, RBCs were incubated for 24 h in the absence (control) or in the presence of HNE in the range between 25 and 50 µM and Ca^++^ endocellular levels were evaluated and expressed as % of positive cells. As shown in Figure 5, HNE determined an increase of intracellular Ca^++^ with values rising from a control value of 1.8% ± 0.2% to 20.5% ± 0.4% and 41.5% ± 0.4% with 25 and 50 µM HNE, respectively (*p* < 0.05).

To confirm the role of Ca^++^ in the eryptotic process, RBCs were incubated with HNE either in the absence or in the presence of Ca^++^ and PS externalization was evaluated. As expected, absence of Ca^++^ resulted in a significant reduction of PS externalization, induced by HNE at 25 or 50 µM, by 68% ± 1% and 72% ± 1% respectively (*p* < 0.05, Figure 6).

As a whole, these results demonstrate that the HNE-induced eryptosis is mediated by a disruption of Ca^++^ homeostasis and are in line with current literature that gives to Ca^++^ a key role in the eryptotic process [1,2,4,37,47]. The mechanism responsible for the increased Ca^2+^ levels might well be related to the opening of Ca^2+^-permeable NSCM channels as it happens with other already-known proeryptotic stimuli [3]. While the molecular identity of these transporters remains elusive, they might partially involve the transient receptor potential channel TRPC6. The lack of TRPC6, however, does not completely abrogate Ca^2+^ entry, suggesting that further cation channels are operating in erythrocytes. The structure of those channels remains to be established [48]. Finally, consistent with our findings, other reports described Ca^++^ as one the signalling tools that HNE uses to exert both its pathophysiological and proapoptotic effects [10,19,20,27,49].

### 3.6. Endocellular Redox Variations Are Not Involved in the HNE-Induced Eryptosis in Human RBCs

RBCs are considered “reporter cells” for the oxidative state since they can be particularly vulnerable to RONS due to their constant exposure to by-products generated by autoxidation of haemoglobin [50]. Moreover, oxidative stress has been associated in phospholipid remodeling and cell dysfunction in a number of eryptotic processes associated with several clinical conditions such as heart failure, diabetes, anaemia, and chronic inflammatory diseases [2,3,4,5,6]. Oxidative stress is, indeed, effective at least in part by activating Ca^2+^-permeable NSCM channels and by initiating a series of events ultimately leading to cell shrinkage and PS externalization [51]. In light of our present results underlying the key role exerted by Ca^2+^ in our experimental system, we next evaluated the involvement of oxidative stress in HNE-induced eryptosis. To this end, RBCs were incubated for 24 h in the absence (control) or in the presence of HNE between 25 and 50 µM and both endocellular RONS and GSH/GSSG levels evaluated. As shown in Figure 7, HNE did not induce any variation of intracellular levels of both RONS and GSH/GSSG at all the concentration tested.

Notwithstanding the role of oxidative stress in eryptosis induced by other agents [1,3,52] and in the HNE-induced apoptosis or cytotoxicity [20,26,53], our results rule out that significant endocellular redox modifications are involved in the Ca^++^-dependent, HNE-induced eryptosis. Other mechanisms, related to NSCM activation, could therefore underlie the process.

### 3.7. PGHS Activation and PGE_2_ Levels Are Involved in the HNE-Induced Eryptosis in Human RBCs

PGHS and PGE_2_ are deeply involved in the eryptotic signalling induced by several stimuli, through the opening of NSCM channels that leads to an increase of cytosolic Ca^++^ levels. Along these lines, we therefore evaluated whether the Ca^++^ influx in HNE-treated RBCs was dependent by activation of PGHS. To this end, RBCs were preincubated either in the absence or in the presence of 50 µM ASA and stimulated with HNE in the range between 25 and 50 µM, and then, both endocellular Ca^++^ levels and PS externalization were evaluated. As shown in Figure 8, with respect to RBCs stimulated with HNE at 25 and 50 µM, pretreatment with ASA resulted in a reduction of PS externalization by 55% ± 1% and by 68% ± 2%, respectively; similarly, Ca^++^ release was reduced by −71% ± 2% and −69% ± 1%, respectively.

In light of these results, we then assessed the effects of HNE on PGE_2_ production. To this end, RBC were incubated for 24 h in the absence (control) or in the presence of HNE in the range between 25 and 50 µM and PGE_2_ release was evaluated. When compared to control cells, treatment of RBCs with HNE significantly modified endocellular PGE_2_ levels that increased from a control value of 5.1 ± 0.1 to 10.1 ± 0.3 and 14.6 ± 0.3 pg/10^8^ RBCs with 25 and 50 µM HNE, respectively (*p* < 0.05, Figure 9).

As a whole, these results demonstrate that the HNE-induced eryptosis is mediated by PGHS activation (essential to rise of Ca^++^ levels and PS externalization) and accompanied by the resulting increase of PGE_2_ levels. While ruling out a direct interaction of HNE with NSCM channels, our data also implied an HNE-induced synthesis of PGE_2_ in the RBCs. The aldehyde, therefore, seems to share with other proeryptotic stimuli a common PGE_2_-dependent mechanism to promote an increase of Ca^++^ levels [1,3,4,54]. Finally, focusing on the HNE-induced PGHS modulation, it is worth to underline that aldehyde has widely been reported to modulate AA metabolism by upregulating COX-2 protein levels in numerous experimental systems relevant to several diseases (atherosclerosis, osteoarthritis, obesity, and inflammation) [55,56,57,58,59]. Our results, obtained in the anucleate RBCs, indicate for HNE a further mechanism through which the aldehyde can stimulate AA metabolism that proceeds via PGHS activation rather than via its upregulation.

### 3.8. Increased Levels of Endocellular Ceramide Are Involved in the HNE-Induced Eryptosis in Human RBCs

Lang at al. proved that two signalling pathways converge to trigger eryptosis. In the first one, PLA_2_ activation produces AA, which then transforms into PGE_2_ that stimulates the entrance of Ca^++^ into the cell [54]. In the second pathway, PLA_2_ activation also produces a liso-derivate which transforms into PAF that activates SMase. This latter ends up promoting PS externalization and eryptosis by generating ceramides [47]. Along these lines, several xenobiotics and numerous proeryptotic conditions (hyperosmotic shock, fever, sepsis, HUS, uraemia, hepatic failure, and Wilson’s disease) have been reported to stimulate eryptosis at least in part by increasing ceramide abundance [52,60].

Along these lines, having observed the key role of Ca^++^ in our experimental system, we then focused on the second mechanism and evaluated the involvement of ceramide in the HNE-induced eryptosis. To this end, RBCs were incubated for 24 h in the absence (control) or in the presence of HNE in the range between 25 and 50 µM and then ceramide levels were assessed as % positive cells. Interestingly, treatment of RBCs with HNE significantly modified the percentage of RBCs containing ceramide that increased from a control value of 5.4 ± 0.1 to 20.5 ± 0.3 and 44.3 ± 0.3 with 25 and 50 µM HNE respectively (*p* < 0.05, Figure 10).

As a whole, our results demonstrate that HNE at physiopathological concentrations induces eryptosis by generating an increase of both PGE_2_ and ceramide. Along these lines, both these signals, produced as consequence of PLA_2_ activation, might result from an HNE-induced alteration of membrane structure.

### 3.9. HNE-Induced Eryptosis in Human RBCs is Associated with Endothelial Cell Adhesion and Dysfunction

Physiological PS externalization is crucial for injured RBCs to be cleared up by macrophages and to prevent premature haemolysis and the following release of haemoglobin in the circulation. On the other hand, through PS exposure, eryptotic RBCs can adhere to ECs and can severely dysfunctionate vascular wall in several pathological conditions such as sickle cell anaemia, malaria infections, diabetes, polycythaemia vera, and retinal vein occlusion [6,61,62].

In line with this, we finally investigated to what extent HNE-induced eryptosis would increase RBC adherence to ECs. To this end, confluent HUVEC were co-cultured for 48 h either with control RBCs or with HNE-stimulated RBCs in the range between 25 and 50 µM. Adherence of RBCs to ECs was evaluated from the ratio of Hb of lysed HUVEC to Hb of RBCs applied to well. As shown in Figure 11A, HNE treatment induced a significant increase of RBC adherence to ECs, with a value increasing from 0.1% ± 0.01% with control cells to 1.5% ± 0.3% and 3.3% ± 0.2% with HNE-treated RBCs at 25 and 50 µM, respectively (*p* < 0.05).

RBC adherence to ECs is a physiopathological process eventually leading to endothelial activation and dysfunction, i.e., overexpression of endothelial cell-surface adhesion molecules, such as ICAM-1 [63]. Induced by pro-inflammatory cytokines, these proteins are required for the recruitment and attachment of inflammatory cells across the endothelium of post-capillary venules.

In order to assess whether HNE-induced eryptosis could have also induced EC dysfunction, we finally assessed ICAM-1 levels in ECs co-incubated for 24 h with either control RBCs or with HNE-stimulated RBCs in the range between 25 and 50 µM. As shown in Figure 11B, HNE induced a significant increase of ICAM-1 levels with a value increasing from 10 ± 1 MFI with control RBCs to 51 ± 3 MFI and 94 ± 1 MFI HNE-treated RBCs at 25 and 50 µM, respectively (*p* < 0.05).

Relevantly, our results demonstrate that HNE-induced eryptosis could foster adhesion of eryptotic RBCs to endothelium and, importantly, lead to its dysfunction. Several pathological conditions, characterized by high plasma levels of HNE, are associated with vascular complications. Our experimental evidences suggest that eryptotic RBCs could contribute to vascular dysfunctions in these diseases.

## 4. Conclusions

In this work, we unveiled for the first time a new physiopathological role for HNE, i.e., its ability to exert eryptotic effects. Mechanistic details of the process pointed out a crucial role for lipid-derived signalling molecules such as ceramide and PGE_2_ (Scheme 1). SEM analysis demonstrated an altered morphology characteristic of apoptotic RBCs and agglutination elements through which the cells could interact with platelets or ECs. Indeed, we have also shown an increased adhesiveness of eryptotic RBCs to endothelial cells and their consequential dysfunction. As a whole, our results may suggest a new and additional mechanism through which HNE could contribute to the vascular complications characteristic of cardiometabolic and inflammatory-mediated pathologies.

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
