# Peer review of "Proeryptotic Activity of 4-Hydroxynonenal: A New Potential Physiopathological Role for Lipid Peroxidation Products"

_biomolecules, 2020, doi:10.3390/biom10050770_

Round 1

Reviewer 1 Report

In this study, the authors have investigated the role of 4-HNE in eryptosis and the underlying mechanisms. Briefly, the authors found that 4-HNE exposure led to eryptosis in a concentration-dependent manner which involved morphological alterations and elevated calcium levels. They claimed that this process is independent of redox modifications and rather dependent on endocellular ceramides. Overall, this is an excellent study. Experiments are well planned, and the manuscript is well written overall.  The current findings will be of interest to a wide range of readers including scientists and clinicians. The data are convincing with appropriate statistical analysis and support their overall findings. However, I have a few concerns/comments for the authors to consider and some additional data are required to support their claims. Addressing these concerns will further improve this manuscript. My specific comments are below:

  1. Figures 1 & 2 can be combined into one section.
  2. Significance can be denoted using special characters (* or #) for simplification and understanding.
  3. Although the FSC data in figure is helpful, the authors should provide specific data related to the effects of 4-HNE on average cell diameter or volume. Again, figures 3 and 4 can be combined.
  4. In figure 6, the authors stated, “Image is representative of six independent experiments carried out in triplicate.” Did the authors perform 18 western blots for the caspase-3 and procaspase-3 data? It is not clear as to why this many sample numbers/data points are required to achieve such dramatic significance?
  5. This study shows that 4-HNE induced eryptosis involves an increase in intracellular calcium levels. The authors should highlight the potential mechanisms related to elevated calcium levels in RBCs in the discussion/conclusion section. Does this involve TRP or other calcium channels as shown previously in other studies (https://doi.org/10.1016/j.freeradbiomed.2016.09.021 and https://doi.org/10.1073/pnas.1600371113) .
  6. Authors should provide data on reactive oxygen species (ROS) levels in figure 9, as 4-HNE exposure may lead to ROS production instead of RNS production. Therefore, measurements of ROS would be more appropriate.
  7. Oxidative stress or reactive aldehyde exposure often leads to glutathione oxidation while maintaining a balance between GSH synthesis and total glutathione. Therefore, measuring the ratio of reduced to oxidized glutathione would more accurate and necessary to conclude that these data rule out the possibility of redox modification in 4-HNE-induced eryptosis. The authors should consider measuring intracellular GSH/GSSG levels to demonstrate this.
  8. The authors have done an excellent job in detailing the mechanism involved in 4-HNE mediated eryptosis. A schematic representation of the proposed mechanisms would be helpful and will further improve the manuscript.
  9. The authors should check for typo errors. Ex: Figure 3

Author Response

                                                                                                       Prof. Neven Zarkovic

Guest Editor of the Special Issue:

“Lipid Metabolism in Health and Disease”

Biomolecules

Dear Prof. Zarkovic,

I have the pleasure of uploading the revised version of the manuscript entitled “Proeryptotic activity of 4-Hydroxynonenal: a new potential physiopathological role for lipid peroxidation products.” by M. Allegra, I. Restivo, A. Fucarino, A. Pitruzzella, S. Vasto, M.A. Livrea, L. Tesoriere and A. Attanzio, according to the reviewer’s suggestions.

Please find below attached our responses to the reviewers' comments:

Reviewer 1

  1. Figures 1 & 2 can be combined into one section.

We thank the reviewer for his/her suggestion.

We combined Figures 1&2.

  1. Significance can be denoted using special characters (* or #) for simplification and understanding.

Again, thanks for this suggestion. We amended all figures and legends accordingly.

  1. Although the FSC data in figure is helpful, the authors should provide specific data related to the effects of 4-HNE on average cell diameter or volume. Again, figures 3 and 4 can be combined.

To the best of our knowledge FCS data are widely utilized in studies focused on Eryptosis. Unfortunately, our SEM apparatus cannot give specific data on cell diameter or volume.

We can either leave the paper as it is or eliminate FSC data and maintain the most relevant part of these results i.e. the morphological modifications as assessed by SEM showing the transition from normal RBC to acanthocytes, echinocytes and polyhedrocytes and the presence of agglutination between eryptotic RBCs with fusion elements.

Figures 3 and 4 have been combined.

  1. In figure 6, the authors stated, “Image is representative of six independent experiments carried out in triplicate.” Did the authors perform 18 western blots for the caspase-3 and procaspase-3 data? It is not clear as to why this many sample numbers/data points are required to achieve such dramatic significance?

We apologise for the typing error. Three western blotting experiments have been performed. Figure legend has been amended accordingly.

  1. This study shows that 4-HNE induced eryptosis involves an increase in intracellular calcium levels. The authors should highlight the potential mechanisms related to elevated calcium levels in RBCs in the discussion/conclusion section. Does this involve TRP or other calcium channels as shown previously in other studies (https://doi.org/10.1016/j.freeradbiomed.2016.09.021 and https://doi.org/10.1073/pnas.1600371113).

We thank the reviewer for this interesting suggestion. The mechanism responsible for the increased Ca2+ levels might well be related to the opening of Ca2+-permeable NSCM channels as it happens with others, already-known proeryptotic stimuli. While the molecular identity of these transporters remains elusive, they might partially involve the transient receptor potential channel TRPC6. The lack of TRPC6, however, does not completely abrogate Ca2+ entry suggesting that further cation channels are operating in erythrocytes. The structure of those channels remains to be established. We agree this information is worth to be considered and we add it in the results/discussion section, as requested (lines 386-391).

  1. Authors should provide data on reactive oxygen species (ROS) levels in figure 9, as 4-HNE exposure may lead to ROS production instead of RNS production. Therefore, measurements of ROS would be more appropriate.

Dichloro-dihydro-fluorescein diacetate is reactive toward a broad range of oxidizing species including Reactive Oxygen and Nitrogen Species (RONS). We understand the reviewer’s perplexity that may have been caused because we have not explained the abbreviation of the term RONS. We apologize for the confusion this may have caused and we have corrected the manuscript by adding this relevant information (line 125).

  1. Oxidative stress or reactive aldehyde exposure often leads to glutathione oxidation while maintaining a balance between GSH synthesis and total glutathione. Therefore, measuring the ratio of reduced to oxidized glutathione would more accurate and necessary to conclude that these data rule out the possibility of redox modification in 4-HNE-induced eryptosis. The authors should consider measuring intracellular GSH/GSSG levels to demonstrate this.

Thanks again to the reviewer for this relevant suggestion. We eliminated the flow-cytometric determination of GSH and performed new experiments employing a colorimetric assay to evaluate intracellular GSH/GSSG levels. Manuscript was amended accordingly (Figure 7 and lines 181-183).

  1. The authors have done an excellent job in detailing the mechanism involved in 4-HNE mediated eryptosis. A schematic representation of the proposed mechanisms would be helpful and will further improve the manuscript.

We thank the reviewer for his/her appreciation. We have drawn a schematic representation for the proposed mechanism.

  1. The authors should check for typo errors. Ex: Figure 3

Type errors have been checked. Thanks a lot.

Reviewer 2

  1. Minor editorial remark for p. 72-74: please mention here if the HNE concentrations are relevant for blood plasma because it is missing.

We thank the reviewer and we added this relevant information that was missing (Lines 72, 74, 80-81).

  1. At r. 80 plasma concentration of HNE would be useful compare to the previously added values.

This relevant information has been added.

  1. Materials and methods subchapter is also complete, but there are some points, which require modification or explanation.
  2. RONS – please explain the abbreviation at first appearance

Explained (line 125).

  1. Please explain the cause to measure haemolysis at 408 nm, because absorption maximum of haemoglobin is different, and depends on its oxidation state.

Spectrophotometric analysis of hemoglobin was performed at 408 nm consistent to hemoglobin soret band. The analysis looked for the level of Hb released by possible cell hemolysis and did not investigate the oxidation state of the protein 

  1. The number of samples and replicates at all determinations should be added because there were five blood samples, but here there are six independent experiments, which is not clear.

As described in material and methods section, we performed 6 independent experiments with blood withdrawn from 5 blood donors.

  1. Results and Discussion
  2. Hb release – if the determination was at 408 nm, the results are questionable due to the different absorption spectra of Hb

Please, see the response to the reviewers' comment n. 3. b

Palermo, 05.05.2020

Sincerely

                                                                                                                                Prof. Luisa Tesoriere

Reviewer 2 Report

The topic of the paper is interesting and add some novel insight about the effect of 4-HNE, as one of the main n-6 PUFA peroxidation product, on eryptotis.

The paper is well written, and the reviewer has no, a major concern about the paper.

The introduction contains all of the relevant and up-to-date information about eryptosis, and its provoking factors, also its molecular mechanism.

Minor editorial remark for p. 72-74: please mention here if the HNE concentrations are relevant for blood plasma because it is missing. At r. 80 plasma concentration of HNE would be useful compare to the previously added values.

Materials and methods subchapter is also complete, but there are some points, which require modification or explanation.

  1. 115. RONS – please explain the abbreviation at first appearance
  2. 142. Please explain the cause to measure haemolysis at 408 nm, because absorption maximum of haemoglobin is different, and depends on its oxidation state
  3. 204-208. The number of samples and replicates at all determinations should be added because there were five blood samples, but here there are six independent experiments, which is not clear.

Results and Discussion

The results are well presented, and the discussion is acceptable.

Only one result is questionable:

  1. 236. Hb release – if the determination was at 408 nm, the results are questionable due to the different absorption spectra of Hb

Author Response

(The authors gave the same response as above.)

Round 2

Reviewer 1 Report

The authors have addressed my concerns. The manuscript is ready for publication.